# Learning Conditional Deformable Templates with Convolutional Networks

**Adrian V. Dalca**
CSAIL, MIT
MGH, HMS
adalca@mit.edu

**Marianne Rakic**
D-ITET, ETH
CSAIL, MIT
mrakic@mit.edu

**John Guttag**
CSAIL, MIT

guttag@mit.edu

**Mert R. Sabuncu**
ECE and BME, Cornell

msabuncu@cornell.edu

## Abstract

We develop a learning framework for building deformable templates, which play a fundamental role in many image analysis and computational anatomy tasks. Conventional methods for template creation and image alignment to the template have undergone decades of rich technical development. In these frameworks, templates are constructed using an iterative process of template estimation and alignment, which is often computationally very expensive. Due in part to this shortcoming, most methods compute a single template for the entire population of images, or a few templates for specific sub-groups of the data. In this work, we present a probabilistic model and efficient learning strategy that yields either universal or *conditional* templates, jointly with a neural network that provides efficient alignment of the images to these templates. We demonstrate the usefulness of this method on a variety of domains, with a special focus on neuroimaging. This is particularly useful for clinical applications where a pre-existing template does not exist, or creating a new one with traditional methods can be prohibitively expensive. Our code and atlases are available online as part of the VoxelMorph library at http://voxelmorph.csail.mit.edu.

## 1 Introduction

A deformable template is an image that can be geometrically deformed to match images in a dataset, providing a common reference frame. Templates are a powerful tool that enables the analysis of geometric variability. They have been used in computer vision [26, 37, 42], medical image analysis [3, 21, 40, 50], graphics [44, 66], and time series signals [1, 73]. We are motivated by the study of anatomical variability in neuroimaging, where collections of scans are mapped to a common template with anatomical and/or functional landmarks. However, the methods developed here are applicable to other domains.

Analysis with a deformable template is often done by computing a smooth deformation field that *aligns* the template to another image. The deformation field can be used to derive a measure of the differences between the two images. Rapidly obtaining this field to a given template is by itself a challenging task and the focus of extensive research.

A template can be chosen as one of the images in a given dataset, but often these do not represent the structural variability and complexity in the image collection, and can lead to biased and misleading analyses [40]. If the template does not adequately represent dataset variability, such as the possible anatomy, it becomes challenging to accurately deform the template to some images. A good template therefore minimizes the geometric distance to all images in a dataset. There has been extensive methodological development for finding such a central template [3, 21, 40, 50], but these methods involve costly optimization procedures and domain-specific heuristics, requiring extensive runtimes. For complex 3D images such as MRI, this process can consume days to weeks. In practice, this leads to few templates being constructed, and researchers often use templates that are not optimal for their dataset. Our work makes it easy and computationally efficient to generate deformable templates.

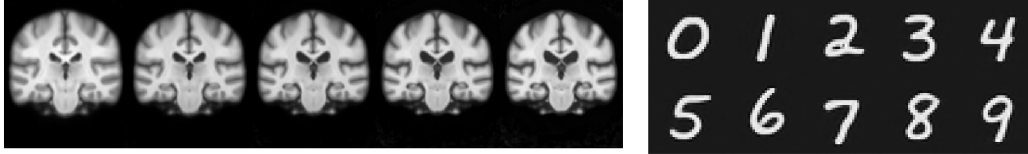

Figure 1: Conditional deformable templates generated by our method. Left: slices from 3D brain templates conditioned on age; Right: MNIST templates conditioned on class label.

While deformable templates are powerful, a single template may be inadequate at capturing the variability in a large dataset. Existing methods alleviate this problem by grouping subpopulations, usually along a single attribute, and computing separate templates for each group. This approach relies on arbitrary decisions about the attributes and thresholds used for subdividing the dataset. Furthermore, each template is only constructed based on a subset of the data, thus exploiting fewer images, leading to sub-optimal templates. Instead, we propose a learning-based approach that can compute on-demand *conditional* deformable templates by leveraging the entire collection. Our framework enables the use of multiple attributes, continuous (e.g., age) or discrete (e.g., sex), to condition the template on, without needing to apply arbitrary thresholding or subdividing a dataset.

We formulate template estimation as a learning problem and describe a novel method to tackle it.

(1) We describe a probabilistic spatial deformation model based on diffeomorphisms. We then develop a general, end-to-end framework using convolutional neural networks that jointly synthesizes templates and rapidly provides the deformation field to any new image.

(2) This framework also enables learning a *conditional* template function given instance attributes, such as the age and sex of the subject in an MRI. Once learned, this function enables rapid synthesis of on-demand conditional templates. For example, it could construct a 3D brain MRI template for 35 year old women.

(3) We demonstrate the template construction method and its utility on a variety of datasets, including a large neuroimaging study. In addition, we show preliminary experiments indicating characteristics and interesting results of the model. For example, this formulation can be extended to learn image representations up to a deformation.

Conditional templates capture important trends related to attributes, and are useful for dealing with confounders. For example, in studying disease impact, for some tasks it may be helpful to register scans to age-specific templates rather than one covering a wide age range.

## 2 Related Works

### 2.1 Spatial Alignment (Image Registration)

Spatial alignment, or registration, between two images is a building block for estimation of deformable templates. Alignment usually involves two steps: a global affine transformation, and a deformable transformation (as in many optical flow applications). In this work we focus on, and make use of, deformable transformations.

There is extensive work in deformable image registration methods [5, 6, 7, 10, 19, 28, 68, 72, 74]. Conventional frameworks optimize a regularized dense deformation field that matches one image with the other [7, 68]. Diffeomorphic transforms are toplogy preserving and invertible, and have been widely used in computational neuroanatomy analysis [6, 5, 10, 13, 14, 32, 41, 55, 59, 70, 74]. While extensively studied, conventional registration algorithms require an optimization for every pair of images, leading to long runtimes in practice.

Recently proposed learning based registration methods offer a significant speed-up at test time [8, 9, 12, 17, 18, 23, 47, 46, 61, 65, 71]. These methods learn a network that computes the deformation field, either in a supervised (using ground truth deformations), unsupervised (using classical energy functions), or semi-supervised setting. These algorithms have been used for registering an image to an *existing* template. However, in many realistic scenarios, a template is not readily available, for example in a clinical study that uses a specific scan protocol. We build on these ideas in our learning strategy, but jointly estimate a registration network and a conditional deformable template in an unsupervised setting. In parallel, independent work, Weber et al. [64] propose a learning-based

framework for diffeomorphic joint temporal alignment of time-series data called DTAN. DTAN generalizes to test data, outperfoming other joint alignment tools for time-series tasks.

Optical flow methods are closely related to image registration, finding a dense displacement field for a pair of 2D images. Similar to registration, classical approaches solve an optimization problem, often using variational methods [11, 35, 67]. Learning-based optical flow methods use convolutional neural networks to learn the dense displacement fields [2, 25, 36, 38, 60, 69].

## 2.2 Template Construction

Deformable templates, or *atlases*, are widely used in computational anatomy. Specifically, the deformation fields from this template to individual images are often carefully analyzed to understand population variability. The template is usually constructed through an iterative procedure based on a collection of images or volumes. First, an initial template is chosen, such as an example image or a pixel-wise average across all images. Next, all images are aligned (registered) to this template, a better template is estimated from aligned images through averaging, and the process is iterated until convergence [3, 21, 40, 50, 63]. Since the above procedure requires many iterations involving many costly (3D) pairwise registrations, atlas construction runtimes are often prohibitive.

A single population template can be insufficient at capturing complex variability. Current methods often subdivide the population to build multiple atlases. For example, in neuroimaging, some methods build different templates for different age groups, requiring rigid discretization of the population and prohibiting each template from using all information across the collection. Images can also be clustered and a template optimized for each cluster, requiring a pre-set number of clusters [63]. Specialized methods have also been developed that tackle a particular variability of interest. For example, spatiotemporal brain templates have been developed using specialized registration pipelines and explicit modelling of brain degeneration with time [22, 31, 48], requiring significant domain knowledge, manual anatomical segmentations, and significant computational resources. We build on the intuitions of these methods, but propose a general framework that can learn *conditional* deformable templates for any given set of attributes. Specifically, our strategy learns a single network that levarges shared information across the entire dataset and can output different templates as a function of sets of attributes, such as age, sex, and disease state. The conditional function learned by our model generates unbiased population templates for a specific configuration of the attributes.

Our model can be used to study the population variation with respect to certain attributes it was trained on, such as age in neuroimaging. In recent literature on deep probabilistic models, several papers find and explore *latent* axes of important variability in the dataset [4, 15, 30, 33, 43, 51]. Our model can also be used to build conditional geometric templates based on such *latent* information, as we show in our experiments. In this case, our model can be seen as learning meaningful image representations up to a geometric deformation. However, in this paper we focus on observed (measured) attributes, with the goal of explicitly capturing variability that is often a source of confounding.

## 3 Methods

We first present a generative model that describes the formation of images through deformations from an unknown conditional template. We describe a learning approach that uses neural networks and diffeomorphic transforms to jointly estimate the global template and a network that rapidly aligns it to each image.

### 3.1 Probabilistic model

Let $x_i$ be a data sample, such as a 2D image, a 3D volume like an MRI scan, or a time series. For the rest of this section, we use images and volumes as an example, but the development applies broadly to many data types. We assume we have a dataset $\mathcal{X} = \{x_i\}$, and model each image as a spatial deformation $\phi_{v_i}$ of a global template $t$. Each transform $\phi_{v_i}$ is parametrized by the random vector $v_i$.

We consider a model of a *conditional* template $t = f_{\theta_t}(a)$, a function of attribute vector $a$, parametrized by global parameters $\theta_t$. For example, $a$ can encode a class label or phenotypical information associated with medical scans, such as age and sex. In cases where no such conditioning information is available or of interest, this formulation reduces to a standard single template for the entire dataset: $t = t_{\theta_t}$, where $\theta_t$ can represent the pixel intensity values to be estimated.

We estimate the deformable template parameters $\boldsymbol{\theta}_t$ and the deformation fields for every data point using maximum likelihood. Letting $\mathcal{V} = \{\boldsymbol{v}_i\}$ and $\mathcal{A} = \{\boldsymbol{a}_i\}$,

$$\hat{\boldsymbol{\theta}}_t, \hat{\mathcal{V}} = \arg\max_{\boldsymbol{\theta}_t, \mathcal{V}} \log p_{\boldsymbol{\theta}_t}(\mathcal{V}|\mathcal{X}, \mathcal{A}) = \arg\max_{\boldsymbol{\theta}_t, \mathcal{V}} \log p_{\boldsymbol{\theta}_t}(\mathcal{X}|\mathcal{V}; \mathcal{A}) + \log p(\mathcal{V}), \tag{1}$$

where the first term captures the likelihood of the data and deformations, and the second term controls a prior over the deformation fields.

**Deformations.** While the method described in this paper applies to a range of deformation parametrization $\boldsymbol{v}$, we focus on diffeomorphisms. Diffeomorphic deformations are invertible and differentiable, thus preserving topology. Specifically, we treat $\boldsymbol{v}$ as a stationary velocity field [5, 17, 32, 45, 46, 57], although time-varying fields are also possible. In this setup, the deformation field $\boldsymbol{\phi}_v$ is defined through the following ordinary differential equation:

$$\frac{\partial \boldsymbol{\phi}_v^{(t)}}{\partial t} = \boldsymbol{v}(\boldsymbol{\phi}_v^{(t)}), \tag{2}$$

where $\boldsymbol{\phi}^{(0)} = Id$ is the identity transformation and $t$ is time. We can obtain the final deformation field $\boldsymbol{\phi}^{(1)}$ by integrating the stationary velocity field $\boldsymbol{v}$ over $t = [0, 1]$. We compute this integration through *scaling and squaring*, which has been shown to be efficiently implementable in automatic differentiation platforms [18, 45].

We model the velocity field prior $p(\mathcal{V})$ to encourage desirable deformation properties. Specifically, we first assume that deformations are smooth, for example to maintain anatomical consistency. Second, we assume that population templates are unbiased, restricting deformation statistics. Letting $\boldsymbol{u}_v$ be the spatial displacement for $\boldsymbol{\phi}_v = Id + \boldsymbol{u}_v$, and $\nabla \boldsymbol{u}_i$ be its spatial gradient,

$$p(\mathcal{V}) \propto \exp\{-\gamma\|\bar{\boldsymbol{u}}_v\|^2\} \prod_i \mathcal{N}(\boldsymbol{u}_{v_i}; \underline{0}, \boldsymbol{\Sigma}_u) \tag{3}$$

where $\mathcal{N}(\cdot; \boldsymbol{\mu}, \boldsymbol{\Sigma})$ is the multivariate normal distribution with mean $\boldsymbol{\mu}$ and covariance $\boldsymbol{\Sigma}$, and $\bar{\boldsymbol{u}}_v = 1/n \sum_i \boldsymbol{u}_{v_i}$. We let $\boldsymbol{\Sigma}^{-1} = \boldsymbol{L}$, where $\boldsymbol{L} = \lambda_d \boldsymbol{D} - \lambda_a \boldsymbol{C}$ is (a relaxation of) the Laplacian of a neighborhood graph defined on the pixel grid, with the graph degree matrix $\boldsymbol{D}$ and the pixel neighbourhood adjacency matrix $\boldsymbol{C}$ [17]. Using this formulation, we obtain

$$\log p(\mathcal{V}) = -\gamma\|\bar{\boldsymbol{u}}\|^2 - \sum_i \frac{d}{2}\lambda_d\|\boldsymbol{u}_i\|^2 + \sum_i \frac{\lambda_a}{2}\|\nabla \boldsymbol{u}_i\|^2 + \text{const} \tag{4}$$

where $d$ is the neighbourhood degree. The first term encourages a small *average* deformation across the dataset, encouraging a central, unbiased template. The second and third terms encourage templates that minimize deformation size and smoothness, respectively, and $\gamma$, $\lambda_d$ and $\lambda_a$ are hyperparameters.

**Data Likelihood.** The data likelihood $p(\boldsymbol{x}_i|\boldsymbol{v}_i, \boldsymbol{a}_i)$ can be adapted to the application domain. For images, we often adopt a simple additive Gaussian model coupled with a deformable template:

$$p(\boldsymbol{x}_i|\boldsymbol{v}_i; \boldsymbol{a}_i) = \mathcal{N}(\boldsymbol{x}_i; f_{\boldsymbol{\theta}_t}(\boldsymbol{a}_i) \circ \boldsymbol{\phi}_{v_i}, \sigma^2\mathbb{I}), \tag{5}$$

where $\circ$ represents a spatial warp, and $\sigma^2$ represents additive image noise. However, in some datasets, different likelihoods are more appropriate. For example, due to the spatial variability of contrast and noise in MRIs, likelihood models that result in normalized cross correlation loss functions have been widely shown to lead to more robust results, and such models can be used with our framework [6].

## 3.2 Neural Network Model

To solve the maximum likelihood formulation (1) given the model instantiations specified above, we design a network $g_{\boldsymbol{\theta}}(\boldsymbol{x}_i, \boldsymbol{a}_i) = (\boldsymbol{v}_i, \boldsymbol{t})$ that takes as input an image and an attribute vector to condition the template on (this could be empty for global templates). The network can be effectively seen as having two functional parts. The first, $g_{t,\boldsymbol{\theta}_t}(\boldsymbol{a}_i) = \boldsymbol{t}$, produces the conditional template. The second, $g_{v,\boldsymbol{\theta}_v}(\boldsymbol{t}, \boldsymbol{x}_i) = \boldsymbol{v}_i$, takes in the template and a data point, and outputs the most likely velocity field (and hence deformation) between them. By learning the optimal parameters $\hat{\boldsymbol{\theta}} = \{\hat{\boldsymbol{\theta}}_t, \hat{\boldsymbol{\theta}}_v\}$, we estimate a global network that simultaneously provides a deformable (conditional) template and its deformation to a datapoint. Figure 2 provides an overview schematic of the proposed network.

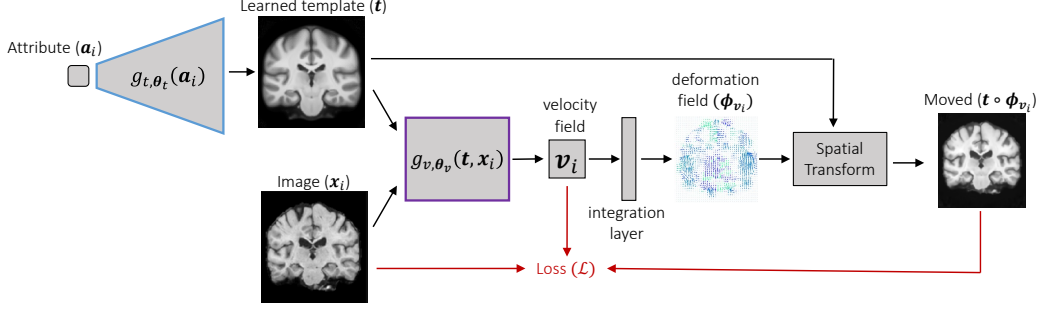

Figure 2: **Overview.** The network takes as input an image and an optional attribute vector. The upper network $g_{t,\theta_t}(\cdot)$ outputs a template, which is then registered with the input image by the second network $g_{v,\theta_v}(\cdot)$. The loss function, derived from the negative log likelihood of the generative model, leverages the template warped into $\boldsymbol{t} \circ \boldsymbol{\phi}_{v_i}$.

We optimize the neural network parameters $\boldsymbol{\theta}$ using stochastic gradient algorithms, and minimize the negative maximum likelihood (1) for image $\boldsymbol{x}_i$:

$$\mathcal{L}(\boldsymbol{\theta}_t, \boldsymbol{\theta}_v; \boldsymbol{v}_i, \boldsymbol{x}_i, \boldsymbol{a}_i) = -\log p_\theta(\boldsymbol{v}_i, \boldsymbol{x}_i; \boldsymbol{a}_i) = -\log p_\theta(\boldsymbol{x}_i | \boldsymbol{v}_i; \boldsymbol{a}_i) - \log p_\theta(\boldsymbol{v}_i)$$

$$= -\frac{1}{2\sigma^2}\|\boldsymbol{x}_i - g_{t,\theta_t}(\boldsymbol{a}_i) \circ \boldsymbol{\phi}_{v_i}\|^2 - \gamma\|\bar{\boldsymbol{u}}\|^2 - \lambda_d \frac{d}{2}\sum_i \|\boldsymbol{u}_i\|^2 + \frac{\lambda_a}{2}\sum_i \|\nabla \boldsymbol{u}_i\|^2 + \text{const}, \quad (6)$$

where $g_{t,\theta_t}(\boldsymbol{a}_i)$ yields the template at iteration $i$, and $\boldsymbol{v}_i = g_{v,\theta_v}(\boldsymbol{t}_{\theta_t,i}, \boldsymbol{x}_i)$.

The use of stochastic gradients to update the networks enables us to learn templates faster than conventional methods by avoiding the need to compute final deformations at each iteration. Intuitively, with every iteration the network learns to output a template, optionally conditioned on the attribute data, that can be smoothly and invertably warped to every image in the dataset.

We implement the template network $g_{t,\theta_t}(\cdot)$ with two versions, depending on whether we are estimating an unconditional or conditional template. The first, conditional version $g_{t,\theta_t}(\boldsymbol{a}_i)$ consists of a decoder that takes as input the attribute data $\boldsymbol{a}_i$, and outputs the template $\boldsymbol{t}$. The decoder includes a fully connected layer, followed by several blocks of upsampling, convolutional, and ReLu activation layers. The second, unconditional version $g_{t,\theta_t}$ has no inputs and simply consists of a learnable parameter at each pixel. The registration network $g_{v,\theta_v}(\boldsymbol{t}, \boldsymbol{x}_i)$ takes as input two images $\boldsymbol{t}$ and $\boldsymbol{x}_i$ and outputs a stationary velocity field $\boldsymbol{v}_i$, and is designed as a convolutional U-Net like architecture [62] with the design used in recent registration literature [9]. To compute the loss (6), we compute the deformation field $\boldsymbol{\phi}_{v_i}$ from $\boldsymbol{v}_i$ using differentiable scaling and squaring integration layers [17, 45], and the warped template $\boldsymbol{t} \circ \boldsymbol{\phi}_{v_i}$ using spatial transform layers. We approximate the average deformation $\bar{\boldsymbol{u}}$ in the loss function using a weighted running average $\bar{\boldsymbol{u}} \sim \sum_{k=K-c}^{K} \boldsymbol{u}_k$, where $\boldsymbol{u}_k$ is the displacement at iteration $k$, $K$ is the current iteration, and $c$ is usually set to 100 in our experiments. Specific network design parameters depend on the application domain, and are included in the supplementary materials.

### 3.3 Test-time Inference of Template and Deformations.

Given a trained network, we obtain a (potentially conditional) template $\hat{\boldsymbol{t}}$ directly from network $g_{t,\theta_t}(\boldsymbol{a}_i)$ by a single forward pass given input $\boldsymbol{a}_i$. For each test input image $\boldsymbol{x}_i$, the deformation fields themselves are often of interest for analysis or prediction. The network also provides the deformation $\hat{\boldsymbol{\phi}}_{\hat{v}_i}$, where $\hat{v}_i = g_{v,\theta_v}(\hat{\boldsymbol{t}}, \boldsymbol{x}_i)$.

Often times, the inverse deformation, which takes the image to the template space, is also desired. Using a stationary velocity field representation, obtaining this inverse deformation $\boldsymbol{\phi}_v^{-1}$ is easy to compute by integrating the negative velocity field using the same scaling and squaring layer: $\boldsymbol{\phi}_v^{-1} = \boldsymbol{\phi}_{-v}$ [5, 18, 56].



Figure 3: **MNIST examples** (1) MNIST digits from `D-scale-rot`; (2) templates conditioned on class (vertical axis) and scale (horizontal axis) on MNIST `D-scale`, learned with our model, and (3) with a decoder-only baseline model; (4) conditional templates learned with our model on the MNIST `D-class-scale-rot` dataset for the digit 3 and a variety of scaling and rotation values.

## 4 Experiments

We present two main sets of experiments. The first set uses image-based datasets MNIST and Google QuickDraw, with the goal of providing a picture of the capabilities of our method. While deformable templates in these data are not a real-world application, these are often-studied datasets that provide a platform to analyze aspects of deformable templates.

In contrast, the second set of experiments is designed to demonstrate the utility of our method on a task of practical importance, analysis of brain MRI. We demonstrate that our method can produce high quality deformable templates in the context of realistic data, and that conditional deformable templates capture important anatomical variability related to age.

### 4.1 Experiment on Benchmark Datasets

**Data**. We use the MNIST dataset, consisting of small 2D images of hand-written digits [49] and 11 classes from the Google QuickDraw dataset [39], a collection of categorized drawings contributed by players in an online drawing game. To evaluate our method's ability to construct conditional templates that accurately capture the impact of attributes on which the templates are conditioned, we generate examples in which the initial images are scaled and rotated (Figure 3). Specifically, we use an image scaling factor in the range $0.7 - 1.3$ and rotations in the range 0 to 360 degrees. We learn different models using either the original dataset involving different classes (`D-class`), the dataset with simulated scale effects (`D-class-scale`), and the rotations (`D-class-scale-rot`). While simulated image changes are obvious to an observer, during training we assume we know the attributes that cause the changes, but do not *a priori* model their effect on the images. This simulates, for example, the correlation between age and changing size of anatomical structures. The goal is to understand whether the proposed method is able to *learn* the relationship between the attribute and the geometrical variability in the dataset, and hence produce a function for generating on-demand templates conditioned on the attributes. The datasets are split into train, validation and test sets.

#### 4.1.1 Validation

In the first experiment, we evaluate our ability to construct suitable conditional templates.

**Hyperparameters.** Model hyperparameters have intuitive effects on the sharpness of templates, the spatial smoothness of registration fields, and the quality of alignments. In practical settings, they should be chosen based on the desired goal of a given task. In these experiments, we tune hyperparameters by visually assessing deformations on validation data, starting from $\gamma = 0.01$, $\lambda_d = 0.001$, $\lambda_a = 0.01$, and $\sigma = 1$ for training on the `D-class` data. We found that once a hyperparameter was chosen for one dataset, only minor tuning was required for other experiments.

**Evaluation criteria**. Template construction is an ill-posed problem, and the utility of resulting templates depends on the desired task. We report a series of measures to capture properties of the resulting templates and deformations. Our first two quantitative evaluation criteria relate to centrality, for which we computed the norm of the mean displacement field $\|\bar{u}\|^2$ and the average displacement size $\frac{1}{n} \sum_i \|u_i\|^2$. Next, we illustrate field regularity per image class, and average intensity image agreement (via MSE). These metrics capture aspects about the deformation fields, rather than solely intrinsic properties of the templates. They need to be evaluated together - otherwise, deformation fields can lead to perfectly matching the image and template while being very irregular and geometrically

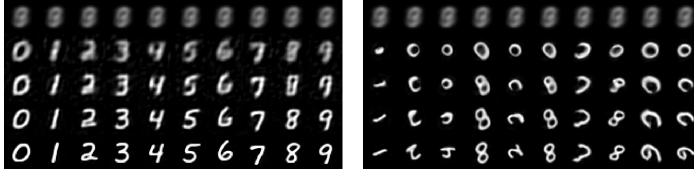 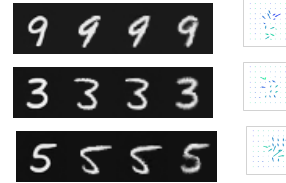

Figure 4: **Example convergence.** Convergence of two conditional template models. Left: model trained on digit-only attribute on `D-class` for epochs $[0, 1, 2, 5, 100]$. Right: model trained on `D-class-rot`, with all three attributes given as input to the model for epochs $[0, 50, 75, 150, 1020]$, and randomly sampled digits $[1, 2, 4, 8, 2, 8, 7, 8, 6, 6]$, rotations, and scales.

Figure 5: **Example deformations.** Each row shows: class template, example class image, template *warped* to this instance, instance *warped* to match the template, and the deformation field.

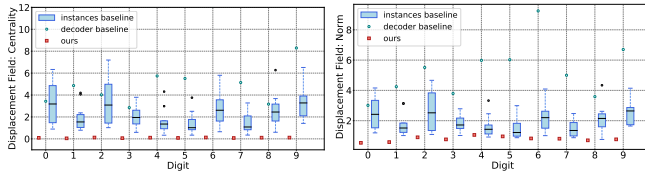 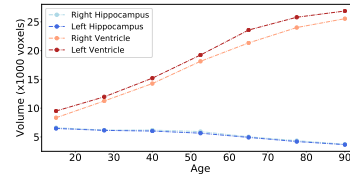

Figure 6: **Quantitative measures.** Centrality and average deformation norm for templates generated by our model and the baselines on the `D-class` variant of MNIST. We find that our models yield more central templates. Additional measures can be found in supplementary Figure 6.

Figure 7: **Volume trends.** Change in volume of ventricles and hippocampi of the age-conditional brain templates.

meaningless, or can be perfectly smooth (zero displacement) at the cost of poor image matching. To capture field regularity, we compute the Jacobian matrix $J_\phi(p) = \nabla\phi(p) \in R^{3\times3}$, which captures the local properties of $\phi$ around voxel pixel $p$. Low values indicate irregular deformation fields, and $|J_\phi(p)| \leq 0$ indicate pixels that are not topology-preserving. Jacobian determinants near $1$ represent very smooth fields. We use held-out test subjects for these measures.

**Baselines**. We compare our templates with templates built by choosing exemplar data as templates, and by training only a decoder of the given attributes using MSE loss and the same network architecture as the template network $g_{t,\theta_t}(\cdot)$. This latter baseline can be seen as differing from our method in that it minimizes a pixel-wise intensity difference as opposed to a geometric difference (deformation).

**Results.** Figure 3 illustrates conditional templates using our model and the decoder, and results from our model on the full MNIST dataset using all attributes. Our method produces sharp, central templates that are plausible digits and are a smoothly deformable to other digits. Example deformations are shown in Figure 5. Supplementary Figures 13 contains similar results for the QuickDraw dataset.

Figure 4 illustrates convergence behavior for two models, showing that the conditional attributes are able to capture complicated geometric differences. Templates early in the learning process share appearance features across attributes, indicating that the network leverages common information across the dataset. The final templates enable significantly smaller deformations than early ones, indicating better representation of the conditional variability. As one would expect, more epochs are necessary for convergence of the model with more attributes.

Figures 6 and 9 show template measures indicating that our conditional templates are more central and require smaller deformations than the baselines when registered with test set digits. We also find that our method and exemplar-based templates can perform well for both deformation metrics, and comparable to each other. Specifically, all deformations are "smooth" (no negative Jacobian determinants) and image differences are visually imperceptible. We underscore that changes in the hyperparameters will produce slightly different trade offs for these measures. At the presented parameters, our method produces templates and deformation fields with slightly smoother deformation fields coming at a slight cost in MSE for some digits, while the baselines can lead to slightly irregular fields to force images to match. The *decoder* baselines underperforms in all metrics. These results indicate that both our methods and instance-based templates can lead to accurate and smooth deformation fields, while our methods produce more central template requiring *smaller* deformations.

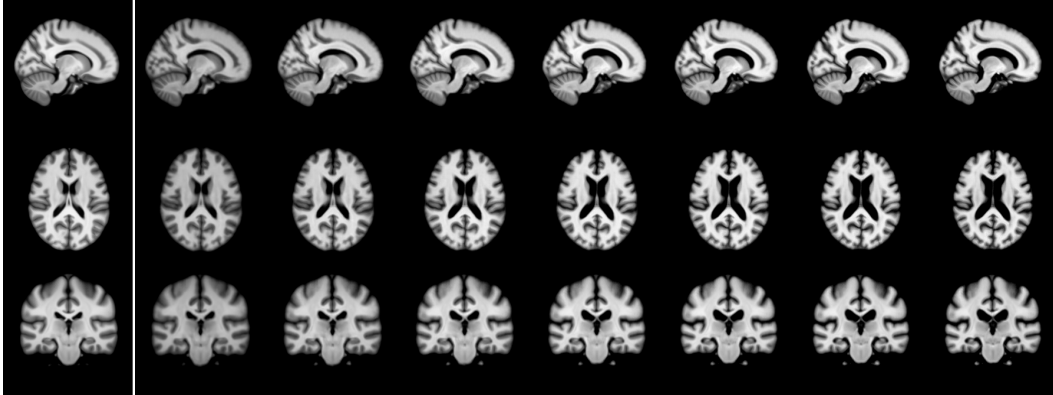

Figure 8: **Slices from Learned 3D Brain MRI templates.** Left: single unconditional template representing the entire population. Right: conditional age templates for brain MRI for ages 15 to 90, illustrating, for example, growth of the ventricles, also evident in a supplementary video.

### 4.1.2 Analysis

In this section, we explore further characteristics and utility provided by our model using the MNIST and QuickDraw dataset. Due to space limitations, the figures are given in the supplementary material.

**Variability and Synthesis.** Conditional deformable templates capture an image representation up to a spatial deformation. Deformation fields from templates to images are often studied to characterize and visualize variability in a population. To illustrate this point, we demonstrate the main within-class variability by finding the principal components of the velocity fields using PCA. Figure 10 illustrates synthesized digits by warping the template along these components capturing handwriting variability in natural digits.

In another variability experiment, we treat *scale* as a confounder and validate that our method reduces confounding effects. Figure 10 illustrates that a model learned with a scale attribute is able to learn principal geometric variability with reduced scale effects compared to one not using this attribute.

**Missing Attributes.** We test the ability of our conditional framework to learn templates that generalize to sparsely observed attributes in two regimes. First, for the D-class-scale dataset, we completely hold out scaling factors in the range $0.9 - 1.1$ for images of digits 3, 4 and 5. In the second regime, we hold out all but 5 instances of the digit 5. Figure 11 indicates that for each regime, our method produces reasonable templates even for the held out attributes, indicating that it leverages the entire dataset in learning the conditioning function.

**Latent attributes.** In this experiment, we compare our method to recent probabilistic models in the situation where attributes are not known *a priori*. To do this, we add an encoder from the input image $x_i$ to the latent attribute, and as a baseline train an autoencoder with the same encoder and decoder architectures as used in our model, and the MSE loss. We train on the D-class dataset with a bottleneck of a single neuron simulating the single unknown attribute. While more powerful autoencoders can lead to better *reconstructions* of the inputs, our goal is to explore the *main* mode of variability captured by each method. As Figure 12 shows, this autoencoder produces much fuzzier looking reconstructions, whereas our approach tends to reproduce the template for the given digit image. This is because the autoencoder learns representations to minimize pixel intensity differences, whereas our approach learns representations that minimize spatial deformations. In other words, out model learns image representations with respect to minimal geometric deformations.

### 4.2 Experiment 2: Neuroimaging

In this section, we illustrate unconditional and conditional 3D brain MRI templates learned by our method, with the goal of showing its utility for the realistic task of neuroimaging analysis. We first show that our method efficiently synthesizes a unconditional population template, comparable to existing ones that require significantly more computation to construct. Secondly, we show that our learned *conditional* template function captures anatomical variability as a function of age.

**Data**. We use a large dataset of 7829 T1-weighted 3D brain MRI scans from publicly available datasets: ADNI [58], OASIS [52], ABIDE [24], ADHD200 [54], MCIC [29], PPMI [53], HABS [16],

and Harvard GSP [34]. All scans are pre-processed with standard steps, including resampling to 1mm isotropic voxels, affine spatial normalization and anatomical segmentations using FreeSurfer [27]. Final images are cropped to $160 \times 192 \times 224$. The segmentation maps are only used for analysis. The dataset is split into 7329 training volumes, 250 validation and 250 test. This dataset was first assembled and used in [20]

**Methods.** All of the training data was used to build an unconditional template. We also learned a conditional template function using age and sex attributes, using only the ADNI and ABIDE datasets which provide this information. Following neuroimaging literature, we use a likelihood model resulting in normalized cross correlation data loss. Training the model requires approximately a day on a Titan XP GPU. However, obtaining a conditional template from a learned network requires less than a second.

**Evaluation.** For a given template, we obtain anatomical segmentations by warping 100 training images to the template and averaging their warped segmentations. For the conditional template, we do this for 7 ages equally spaced between 15 and 90 years old, for both males and females. We first analyze anatomical trends with respect to conditional attributes. We then measured registration accuracy facilitated by each template with the test set via the widely used volume overlap measure Dice (higher is better). To compare volume overlap via the Dice metric, as a baseline we use the atlas and segmentation masks available online from recent literature [8]. To test the volume overlap with anatomical segmentations of test data, we warp each template (unconditional, appropriate age and sex conditional template, and baseline) to each of 100 test subjects, and propagate the template segmentations. We computed the mean Dice score of all subjects and 30 FreeSurfer labels.

**Results.** Figures 8 and 14, and a supplementary video[1], illustrate example slices from the unconditional and conditional 3D templates. The ventricles and hippocampi are known to have significant anatomical variation as a function of age, which can be seen in the images. Figure 7 illustrates their volume measured using our atlases as a function of age, showing the growth of the ventricle volumes and shrinkage of the hippocampus. Figure 15 illustrates representative results.

We find Dice scores of $0.800$ ($\pm 0.110$) for the unconditional template, $0.795$ ($\pm 0.116$) for the conditional model, and $0.731$ ($\pm 0.153$) for the baseline, with this difference roughly consistent for each anatomical structure. We emphasize that these numbers may not be directly compared, since the baseline atlas (and segmentations) were obtained using a different process involving an external dataset and manual labeling, while our template was built with our training images (and their FreeSurfer segmentations to obtain template labels). Nonetheless, these visualizations and analyses are encouraging, suggesting that our method provides anatomical templates for brain MRI that enable brain segmentation.

## 5 Discussion and Conclusion

Deformable templates play an important role in image analysis tasks. In this paper, we present a method for automatically learning such templates from data. Our method is both less labor intensive and computationally more efficient than traditional data-driven methods for learning templates. Moreover, our method can be used to learn a function that can quickly generate templates conditioned upon sets of attributes. It can for example generate a template for the brains of 75 year old women in under a second. To our knowledge, this is the only general method for producing templates conditioned on available attributes.

In a series of experiments on popular image datasets, we demonstrate that our method produces high quality unconditional templates. We show that it can be used to construct conditional templates that account for confounders such as scaling and rotation. In a second set of experiments, we demonstrate the practical utility of our methods by applying it to a large data set of brain MRI images. We show that with about a day of training, we can produce unconditional atlases similar in quality and utility to a widely used atlas that took weeks to produce. We also show that the method can be used to rapidly produce conditional atlases that are consistent with known age-related changes in anatomy.

In the future, we plan to explore downstream consequences of being able to easily and quickly produce conditional templates for medical imaging studies. In addition, we believe that our model can be used for other tasks, such as estimating *unknown* attributes (e.g., age) for a given patient, which would be an interesting direction for further exploration.

## Acknowledgments

This research was funded by NIH grants R01LM012719, R01AG053949, and 1R21AG050122, NSF CAREER 1748377, NSF NeuroNex Grant 1707312, and Wistron Corporation.

## Footnotes

[1]Video can be found at `http://voxelmorph.mit.edu/atlas_creation/`

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
