[Supplementary Material]

## Supplementary Data

## 1   Architectures

We model the conditional template network architecture as a decoder with a dense layer followed by several upsampling and convolutional levels. Class attributes are encoded as one-hot representations, and continuous attributes are encoded as scalars. For toy datasets, we use a dense layer from the input attributes to a $4 \times 4$ image with $k$ features followed by three upsampling levels with two convolution layers each with 16 features each. The value for $k$ is set to $8$ in most situations, and in the latent variable experiment, we avoid over-fitting by using a bottleneck of 1 and $k$ of 2, for both our method and the baseline. Unconditional templates involve a single layer with a learnable parameter at each pixel.

For our conditional 3D neuroimaging template, we use a dense layer to a $80 \times 96 \times 112$ 3D image with $8$ features, followed by a level of upsampling with three convolution layers and $8$ features. All kernels are of size $3$.

We base our design for the registration network on the architecture described in recent learning-based registration frameworks [9]. Specifically, we use a U-Net style architecture with four downsampling and upsampling layers, each involving a convolutional layer with 32 features and 3x3 kernel size. This is followed by two more convolution layers. For baseline templates – instances, and those produced by decoder-based models – we learn a registration network using the same architecture.

## 2   Quantitative Measures

Figure 9: **Quantitative measures.** Top: Centrality and average deformation norm for templates generated by our model and the baselines on the `D-class` variant of MNIST. We find that our models yield more central templates. Bottom: Both MSE and Jacobians determinants measures indicate good deformations for all models.

# 3 Figures for additional benchmark experiments

Below we show the figures for the missing data, the latent attributes, and variability experiments.

Figure 10: **Variability. Left**: Images are synthesized by warping a learned template from the D-class dataset along the main two axes found by applying PCA to test deformation fields. **Right**: Images are synthesized by warping learned template using the D-class-scale along the main two axes found by applying PCA. The first model uses scale as an attribute, learning mostly *other* meaningful geometric deformations. The second model does not use scale as an attribute; consequently both principal components are dominated by scale.

Figure 11: **Missing attributes.** Left: during training, digits 3-5 are not synthesized using scaling $0.9-1.1$. Right: during training, only 5 examples of digit 5 are given. Red boxes highlight templates build with attributes where data was held out.

Figure 12: **Latent attribute results.** The top row shows sample input digits. The middle row shows our reconstruction for those input images, highlighting that the model learns a template for each digit type even when the digit attribute is not explicitly given. The bottom row shows templates built using an auto encoder with a single neuron bottleneck, showing that the main variation captured in this manner encourages small pixel intensity error, rather than the geometric difference minimized by our method.

# 4 Quickdraw result examples

Figure 13: **Quickdraw example templates.** Left: example and learned atlases for the `D-class` QuickDraw dataset, and below variability examples similar to Figure 10-left. Right: templates for different scales and classes learned using D-class-scale simulations.

# 5 Additional neuroimaging results

Below we show additional figures of neuroimaging results, including segmentations for (conditional) atlases, and example deformations.

Figure 14: **Segmentations.** Example segmentations overlayed with different brain views for our unconditional template (left) and conditional templates (right) varying by age.

Figure 15: **Example 3D neuroimaging deformations.** Frames include: coronal slices for age-conditional template, subject scan, warped template onto subject, warped subject onto template (using inverse field), and the first two directions of the 3D forward and inverse warps, and velocity field.

# 6   Supplementary Video

We include a supplementary video at `http://voxelmorph.mit.edu/atlas_creation/`, illustrating our method's ability of synthesizing templates on-demand based on given attributes. Specifically, the video illustrates the brain template conditioned on age, between 15 and 90 years old, also used as the video frame index.