[Reviews · NeurIPS 2019]

Reviewer 1



Originality: I had a difficult time evaluating the innovation of their method. In terms of image classification, it is not very novel. Austerweil and Griffiths (NeurIPS 2010) proposed a similar idea, albeit in a Bayesian nonparametric framework (See for faster inference method Hu et al., 2012 ICML: https://arxiv.org/pdf/1206.6482.pdf). This is just within the Bayesian nonparametrics community, and so it is likely that more sophisticated methods for learning templates and transformations simultaneously exist in the computer vision literature. Their description of relevant previous work on neural image registration methods was too sparse (Lines 62-84) to provide enough information for me to evaluate. Quality: The framework and methods are sensible and well-executed (although I was disappointed with the authors' answers to the Reproducibility Checklist that the authors were not committed to posting their code and data with a final paper). Their evaluation method was a bit weak because there were no comparisons to other methods that they mentioned in the previous work section. Even if it were only feasible for a small subset because it was slow and/or the other work performed better, that would still improve the submission by providing more context for evaluating the method and contribution. Clarity: The paper is very well-written with clear descriptions of their data, analyses, and architecture. Significance: I'm not sure if this would be of interest to researchers beyond the neuroimaging community. It is hard for me to evaluate its significance due to the reasons mentioned above. Response to Author Feedback: Thank you for your thoughtful response to my and the other reviewer criticism. My main concern is that if this is a manuscript focusing on neuroimaging analysis and not meant to be compared or given a proper motivation from the perception/image registration literatures, then why spend so much of the manuscript focusing on it rather than neuroimaging analyses? If it is strong enough for NeurIPS on the neuroimaging work alone, then that would be sufficient for me. If the authors wanted to illustrate its potential, then I believe they should address other solutions to the template deformation problem and why they shouldn't be applied to neuroimaging (instead of their own approach). I do not believe the contributions outside of neuroimaging are strong enough for members of those related communities to be a low-medium or greater significance. Additionally, illustrating model comparisons in the main submission would greatly strengthen it. I had put that as an improvement that would potentially increase my score, but it was left unaddressed in their response.

Reviewer 2



Overall * this presented idea is interesting and well-suited for NIPS * the methods are well explained in general * the experiments are serious Little information is given on training parameters and on the neural networks architecture, hence it is hard to judge how strong and efficient the solution is. major ----- Having multiple templates calls for a (difficult) model selection step. Also, it is likely that conditional templates are lower quality than unconditional ones: this should be investigated and documented. Overall I don't feel very enthusiastic about the idea about multiple templates. Hyperparameters: gamma, lambda_d, lambda_a are apparently set arbitrarily: this is bad and typically limits the use of the method. Fig. 6 is bad quality and labels are not readable It is a pity that examples on digits are far more developed than experiments on T1 images. I would have been much more interested by a serious experiment on brain scans, involving e.g. age prediction. minor ----- Awkward to denote A the Laplacian of the graph, as a stands for the template parameter. l.30 "If the template does not adequately capture the dataset variability" but the variability is captured by the deformation model rather than by the template ? l.59 I feel uncomfortable with assertions such as "For example, in studying disease impact, it is helpful to register scans to age-specific 60 templates rather than one covering a wide age range." is this true ? I think that this really depends on what your question is.

Reviewer 3



Training both the templates and the deformations of a registration procedure end-to-end is novel and very useful. The loss function has been thoroughly explained. The explanation for the deformation prior p(V) could be more clearly motivated. It is not clear to me how deformations at different scales are weighted. For example in the case of fMRI registration, I would expect pretty much all images to be rotated and translated as a whole (which should not be given a strong loss), but also locally (which would incur a larger loss). It is not clear to me how these different transformation penalties would be controlled. Perhaps this is regulated by the lambda parameters, but I could not deduce it from the text. I would like to see some more in-between steps of the model. For example, it would be nice to see velocity maps. There is no promise in the paper that source code will be released. It would be helpful if the authors clarified that. Update: I thank the authors for clarifying code release and the whole-image registration step; I'm looking forward to the additional figures to be shown in the supplement. I do not see any additional concerns after the review phase. I still think this is a good paper.

[Author Response · NeurIPS 2019]

We thank the reviewers for their insightful and constructive feedback. Our main contributions are (1) providing a principled model and learning-based method for building deformable templates, (2) extending this to learn *conditional* templates, and (3) demonstrating its usability in a series of experiments, with a focus application of neuroimaging.

Regarding Reviewer 1 and 3's note about *code release*: we **will release code**, model weights and atlases. Perhaps we misunderstood the reproducibility checklist, we meant that we did not provide code *at submission time*. This is because it may break anonymity since the code is deeply integrated with our registration library that we previously open sourced.

We agree with comments from Reviewers 1 and 2 about the importance of neuroimaging experiments. Due to space constraints, we condensed these experiments in the main text, but expanded them in *Supplemental Sec. 1.4: Neuroimaging Analysis*. Specifically, we computed a baseline template using a small dataset and state-of-the-art method (as Reviewer 1 suggests), and demonstrated significantly improved performance of our method compared to this baseline. We also showed example deformations, and segmentation maps overlayed on conditional and unconditional templates. We will expand the description of these experiments in the supplementary, and add detail to the main text.

**Reviewer 1.** We will release code, please see our explanation above.

We agree that our work is not novel in *image classification*, but that is not the topic of our work. Based on the suggested citations, we believe that perhaps the reviewer is referring to one of the possible downstream applications: learning image representations up to a deformation. However, we emphasize that (1) this is not what our main contribution, and (2) our goal there was simply to illustrate that learning with a deformation-based loss yields *different* representations than other reconstruction losses. We believe that neither of the suggested citations is related to this message.

In our paper, "image registration" is one of several related works sections, to which we devote half a page and 40 citations. Unfortunately, due to space limitations, we could not provide more background about this topic.

We agree that our main application focus is neuroimaging, but we envision several application domains. Because of this, we demonstrate several characteristics and uses of our method in the results section. Please see lines 7-12 above, about experimental results – while condensed in the main text, we expanded the analysis in the supplementary material.

**Reviewer 2**. We provide extended architecture details in the *Supplemental Section 1.1: Architectures*. Due to space limitations, and since the architecture is not our focus, we chose to not include it in the main text.

The MNIST experiments were included to demonstrate our method's potential, but we agree that neuroimaging experiments are more important. Please see lines 7-12 above about extended MRI results in supplementary material.

We will clarify in the paper that our conditional method variant does not provide *multiple* templates, but a single conditional template *function*. For a test subject and their attribute value, this function efficiently provides the appropriate template (e.g., 45-year old female brain), and no template selection is required. One utility of conditional templates is that the deformation fields are more informative, for example by eliminating variability from confounding attributes. The reviewer also asks about the *quality* of conditional templates. We found that compared to unconditional templates, conditional templates have similar texture characteristics, enable comparable registration accuracy (Dice score) in general, and improvements for some age groups. For this rebuttal we compared the data-matching loss term yielded via conditional and unconditional templates, and found no statistical difference ($0.69 \pm 0.03$ for both).

We agree that understanding hyper-parameters is important. As we touch upon in *Supplemental Section 1.2*, the model hyper-parameters have intuitive effects on the sharpness of the templates, the spatial smoothness of the registration fields, and the quality of the alignments. We will clarify that they should be chosen based on the desired goal of a given task. For example, in neuroimaging, hyper-parameters could be determined by maximizing the highest anatomical overlap based on segmentations or landmarks in a validation set, as is often done in image registration.

Based on the review suggestion, we believe our model can also be used to estimate *unknown* attributes, which may require model selection. While outside the scope of the current paper, this is an interesting future direction. Along with emphasizing the expanded MRI results, we will make the minor corrections/clarifications, including enlarging Fig. 6.

**Reviewer 3**. We will release code, please see lines 4-6 above. Affine or rigid alignment (such as the rotations mentioned by the reviewer) are usually easy to solve in a pre-processing step, and are generally not of interest in downstream analysis (e.g., not anatomically meaningful). As described in the MRI pre-processing description, standard affine alignment was performed in all of our datasets before using our method.

As suggested, we will expand the motivation of the velocity field prior. For most applications, we often desire spatially smooth deformations to encourage anatomical consistency, leading to the Laplacian prior choice. In estimating templates that represent an anatomical *mean*, we expect the deformations to be generally small, leading to the first prior term.

We appreciate the suggestion to illustrate intermediate steps. While response space is limited, in the paper we will show intermediate features and velocity fields, which appear similar (but less smooth) to the displacement fields in *Sup. 1.4*.

[Meta-Review · NeurIPS 2019]

As pointed out by Reviewers the proposed model is sound and of interest. Despite the apparent inconsistency between a strong motivation from neuroimging and partial empirical analysis on neuroimaging data, the contribution is of interest for the NeurIPS community and of potential impact for the neuroimaging practitioners.